# Satellite based high resolution mapping of rainfall over Southern Africa

Hanna Meyer[1], Johannes Drönner[2], and Thomas Nauss[1]

[1]Environmental Informatics, Faculty of Geography, Philipps-University Marburg, Deutschhausstr. 10, 35037 Marburg, Germany
[2]Database Research Group, Faculty of Mathematics und Informatics, Philipps-University Marburg, Hans-Meerwein-Str. 6, 35032 Marburg, Germany

*Correspondence to:* Hanna Meyer (hanna.meyer@geo.uni-marburg.de)

**Abstract.** A spatially explicit mapping of rainfall is necessary for Southern Africa for eco-climatological studies or nowcasting but accurate estimates are still a challenging task. This study presents a method to estimate hourly rainfall based on data from the Meteosat Second Generation (MSG) Spinning Enhanced Visible and Infrared Imager (SEVIRI). Rainfall measurements from about 350 weather stations from 2010-2014 served as ground truth for calibration and validation. SEVIRI and weather station data were used to train neural networks that allowed the estimation of rainfall area and rainfall quantities over all times of the day. The results revealed that 60 % of recorded rainfall events were correctly classified by the model (Probability Of Detection, POD). However, the False Alarm Ratio (FAR) was high (0.80), leading to a Heidke Skill Score (HSS) of 0.18. Estimated hourly rainfall quantities were estimated with an average hourly correlation of rho = 0.33 and a Root Mean Square Error (RMSE) of 0.72. The correlation increased with temporal aggregation to 0.52 (daily), 0.67 (weekly) and 0.71 (monthly). The main weakness was the overestimation of rainfall events. The model results were compared to the Integrated Multi-satellitE Retrievals for GPM (IMERG) of the Global Precipitation Measurement (GPM) mission. Despite being a comparably simple approach, the presented MSG based rainfall retrieval outperformed GPM IMERG in terms of rainfall area detection where GPM IMERG had a considerably lower POD. The HSS was not significantly different compared to the MSG based retrieval due to a lower FAR of GPM IMERG. There were no further significant differences between the MSG based retrieval and GPM IMERG in terms of correlation with the observed rainfall quantities. The MSG based retrieval, however, provides rainfall in higher spatial resolution. Though estimate rainfall from satellite data remains challenging especially at high temporal resolutions, this study showed promising results towards improved spatio-temporal estimates of rainfall over Southern Africa.

## 1 Introduction

The dynamics of rainfall play an important role in Southern Africa especially in the arid and semi-arid areas where farming is a main income and the quality of the pastures mainly depends on water availability (Fynn and O'Connor, 2000). Accurate nowcasting of rainfall at high temporal and spatial resolutions is therefore of interest for the farmers in Southern Africa and would help them to assess the carrying capacity of their land. It is of further importance as a baseline product for a variety of environmental research studies as rainfall is a key variable for many ecological and hydrological processes.

Rain gauges are still considered as the most accurate way to measure rainfall. Southern Africa features a network of rain gauges operated by the weather services of the individual countries as well as by a variety of research projects. However, the network does not feature a sufficient density to capture spatially highly variable rainfall dynamics. To obtain spatially explicit data, ground-based radar networks are well established to measure rainfall in other parts of the world (e.g. RADOLAN in Germany, Bartels et al. (2004)). A radar network covering the entire region of Southern Africa, however, is currently not available and the existing radar-based rainfall estimates in South Africa are still afflicted with many uncertainties (IPWG, 2016). A satellite-based monitoring of rainfall is therefore an obvious alternative.

A number of global satellite-derived products have been developed in the last decades (e.g. TRMM, CMORPH, PERSIANN, see review in Kidd and Huffman (2011); Prigent (2010); Thies and Bendix (2011); Kidd et al. (2011); Levizzani et al. (2002)). Since 2014, the latest product from the Global Precipitation Measurement (GPM) mission, as a successor of the Tropical Rainfall Measuring Mission (TRMM), provides the most recent global estimates of precipitation at high spatial and temporal resolutions. It might be expected that the GPM products would feature a high degree of accuracy since the TRMM-3B42 product has been identified as the most accurate retrieval at least for east Africa (Cattani et al., 2016).

In addition to global rainfall retrievals, a number of regionally adapted retrievals were developed in the last decades (Kühnlein et al., 2014b, a; Meyer et al., 2016; Feidas and Giannakos, 2012; Giannakos and Feidas, 2013). Kühnlein et al. (2014a, b) and Meyer et al. (2016) presented a methodology to estimate rainfall from optical Meteosat Second Generation (MSG) Spinning Enhanced Visible and InfraRed Imager (SEVIRI) data for Germany. In this approach, machine learning algorithms were used to relate the spectral properties of MSG to reliable radar data as a ground truth. Though the retrieval showed promising results, such spatially comprehensive ground truth data are lacking for Southern Africa. An adaptation of the retrieval technique to Southern Africa hence requires a model training that relies on sparse weather station data as a ground truth.

This study aims to test the suitability of a MSG and artificial neural network based rainfall retrieval which is regionally trained using rain gauge data to provide spatially explicit estimates of rainfall areas and rainfall quantities for Southern Africa. The suitability of the model is assessed by validation with independent weather station data and comparison to the Integrated Multi-satellitE Retrievals for GPM (IMERG) product.

## 2    Methods

The methodology is divided into a pre-processing of satellite and rain gauge data, model tuning and training including its validation, model estimation and comparison to GPM IMERG (Fig. 1).

### 2.1    Study area

The area of investigation comprises South Africa, Lesotho and Swaziland, Namibia, Botswana, Zimbabwe as well as parts of Mozambique (Fig. 2). Average annual rainfall in Southern Africa roughly follows an aridity gradient from the dry west to the more humid east. With the exceptions of some coastal regions in South Africa, most rain falls during the summer months. In the coastal areas of South Africa, frontal systems cause light rain that may last over several days. The majority of interior areas

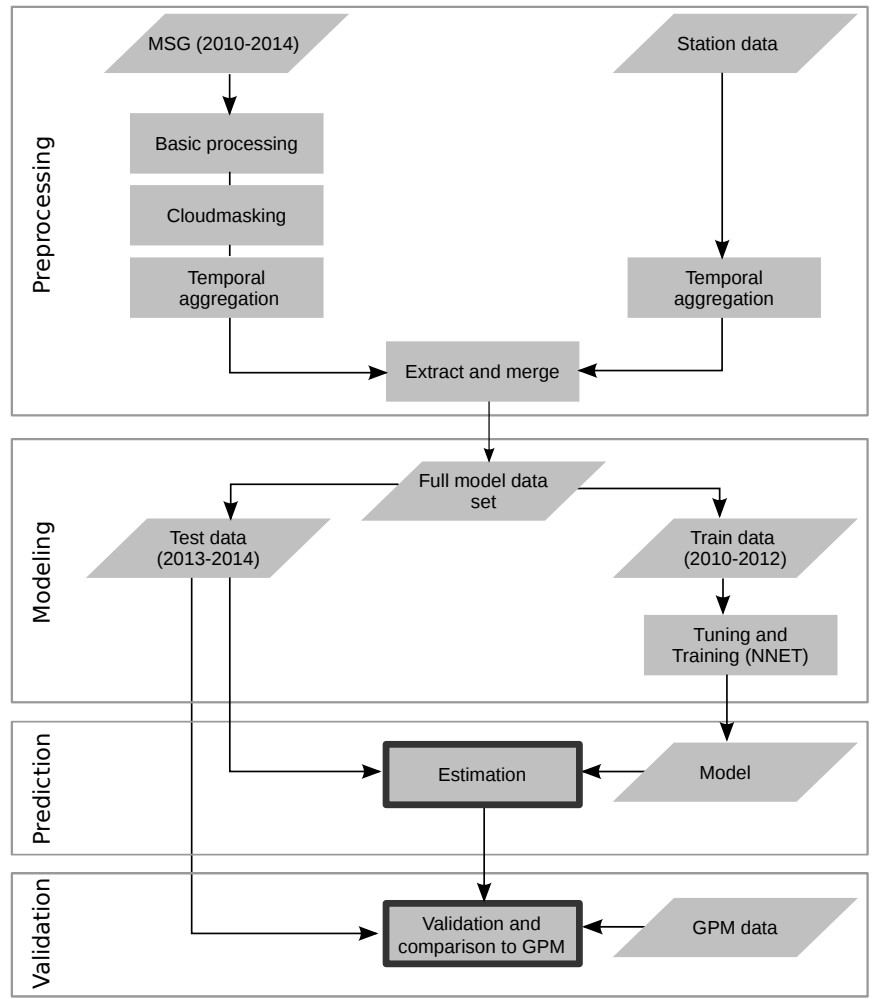

**Figure 1.** Flow chart of the methodology applied in this study.

are dominated by local and short-term convective heavy showers mostly with thunder in the afternoon or evening hours. Rain from synoptic systems lasting up to several days also occurs. Snow and hail only contribute a negligible amount to the overall

precipitation totals. The inter-annual variability of rainfall is high for the arid areas. For a detailed description of Southern African rainfall characteristics see Kruger (2007) and Kaptué et al. (2015).

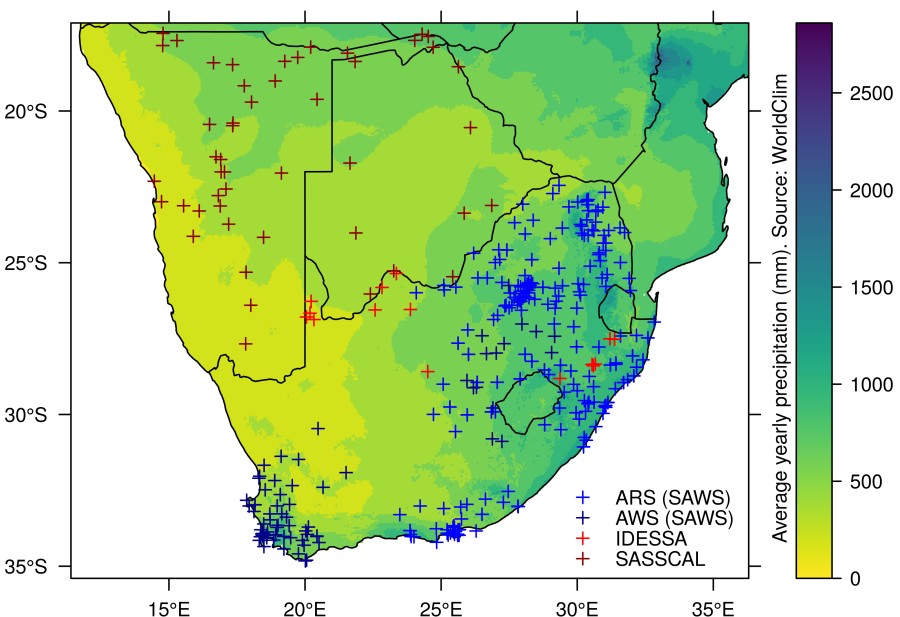

**Figure 2.** Map of the average annual precipitation sums in the study area as estimated by WordClim (Hijmans et al., 2005). Points show the locations of the weather stations that were used as ground truth data in this study. Automatic Rainfall Stations (ARS) and Automatic Weather Stations (AWS) are operated by the South African Weather Service (SAWS). Further stations are operated by SASSCAL WeatherNet as well as by the IDESSA project.

## 2.2 Data and Preprocessing

### 2.2.1 Station data

5   Rainfall data for 2010 to 2014 were obtained from the South African Weather Service (SAWS). The data were recorded at 229 automatic rainfall stations and 91 automatic weather stations (Fig. 2). They were complemented by 22 stations from SASSCAL WeatherNet (www.sasscalweathernet.org/) located in southern Namibia and Botswana. For 2014, data from an additional 15 stations in South Africa operated by the IDESSA project (An Integrative Decision Support System for Sustainable Rangeland Management in Southern African Savannas, www.idessa.org/) were available. The data passed general provider-dependent

10  quality checks before it was used in this study. This includes filtering of data beyond common data ranges, or situational

checks for consistency with related parameters (e.g. air humidity) by SASSCAL. SAWS payed attention to rainfall values $>$ 10 mm within 5 minutes and deleted those values if unreliable. Data from all providers was then included in an on-demand processing database system (Wöllauer et al., 2015) where it was automatically cross-checked for reliability by filtering values $< 0$ and $> 500$ mm of rainfall per hour. All station data that provided sub-hourly information was aggregated to a temporal resolution of 1 hour within the database. Though the station data is not randomly distributed in the model domain, it covers the entire aridity gradient, from sites with very low ($< 200$ mm) precipitation to sites in areas with highest ($\sim 1500$ mm) yearly precipitation sums.

### 2.2.2 Satellite data

MSG SEVIRI (Aminou et al., 1997) scans the full disk every 15 minutes with a spatial resolution of $3 \times 3$ km at sub-satellite point ( $3.5 \times 3.5$ km in Southern Africa). Reflected and emitted radiances are measured by 12 channels, three channels at visible (VIS) and very near infrared wavelengths (NIR, between 0.6 and 1.6 $\mu$m), eight channels ranging from near-infrared to thermal infrared wavelengths (IR, between 3.9 and 14 $\mu$m) and one high-resolution VIS channel with a spatial resolution of $1 \times 1$ km which was not considered in this study.

The rainfall retrieval technique presented here works under the assumption that VIS, NIR and IR channels of MSG SEVIRI provide proxies for microphysical cloud properties, which are, in turn, related to rainfall. VIS and NIR channels have been shown to be related to cloud optical depth (Roebeling et al., 2006; Benas et al., 2017) and cloud water path (Kühnlein et al., 2014b) where the NIR channel is further related to cloud particle size (Roebeling et al., 2006). The IR channels have been shown to provide information about the cloud top temperature which was used as a proxy for cloud height (Hamann et al., 2014). The cloud droplet effective radius as well as liquid water path during night was approximated using IR differences (Merk et al., 2011; Kühnlein et al., 2014b).

MSG SEVIRI Level 1.5 data (EUMETSAT, 2010) was preprocessed to radiance values according to EUMETSAT (2012b) and brightness temperatures according to EUMETSAT (2012a) using a processing scheme based on a custom raster processing extension of the eXtensible and fleXible Java library (see https://github.com/umr-dbs/xxl) which enables parallel raster processing on CPUs and GPUs using OpenCL.

### 2.2.3 Cloud mask

A cloudmask was used to exclude all pixels that were not cloudy in the respective SEVIRI scenes. For 2010 to 2012, the CM SAF CMa Cloudmask product (Kniffka et al., 2014) was applied. Due to the availability of the CM SAF CMa cloudmask dataset which was currently limited to the years 2004 to 2012, we used the cloud mask information of the CLAAS-2 data record (Finkensieper et al., 2016) for the years 2013 and 2014 which is the 2nd edition of the SEVIRI-based cloud property data record provided by the EUMETSAT Satellite Application Facility on Climate Monitoring (CM SAF; see also Stengel et al. (2014) for further information on CLAAS). All pixels that were classified as cloud contaminated or cloud filled were interpreted as cloudy. Pixels that were classified as cloud-free were excluded from further analysis.

## 2.3 Model strategies for rainfall estimation

### 2.3.1 General model framework

The modeling methodology follows the study of Kühnlein et al. (2014a, b) who used the spectral channels of MSG SEVIRI to train a Random Forest model that is able to spatially estimate rainfall areas and rainfall rates over Germany. Based on this study, Meyer et al. (2016) have shown that neural networks outperform the initially used Random Forest algorithm. In these previous studies on the rainfall retrieval, the radar based RADOLAN product (Bartels et al., 2004) was used as ground truths to train the model. The high data quality and spatially explicit information allowed the model to be optimised without too much confusion caused by uncertainties in the training data. However, the goal of the retrieval was that it can be applied to areas where spatially explicit data for rainfall are not available, as it is the case in Southern Africa.

### 2.3.2 Training and test data sets

Cloud masked MSG data from 2010 to 2014 were extracted at the locations of the weather stations. To match the temporal resolution of all available rain gauge data, the extracted data were aggregated to hourly values. This was done by taking the median value of the four scenes available every hour. However, only if all four scenes were masked as cloudy, the corresponding hourly values for a respective station were used for further analysis. The extracted and aggregated MSG data were then matched with the corresponding rain gauge information under consideration of the time shift between MSG data (UTC) and rain gauge data (UTC + 2).

The spectral channels as well as the channel differences $\Delta$ T6.2 - 10.8, $\Delta$ T7.3 - 12.1, $\Delta$ T8.7 - 10.8, $\Delta$ T10.8 - 12.1, $\Delta$ T3.9 - 7.3, $\Delta$ T3.9 - 10.8 and the sun zenith were used as predictor variables during daytime, in accordance to (Kühnlein et al., 2014b) and previous studies on MSG based delineation of cloud properties (see section 2.2.2). Thus, the predictor variables contain the SEVIRI channels as well as channel combinations. Although this partially duplicates information, the channel combinations allow highlighting patterns that might not be apparent in the individual channels. As additional potential predictors, Meyer et al. (2017) tested different cloud texture parameters and have shown that the chosen spectral channels and differences are sufficient as predictors.

Since neural networks require that the predictor variables are standardized, all predictors were centered and scaled by dividing the values of the mean-centered variables by their standard deviations. Since the VIS and NIR channels of MSG are not available during the nighttime, the dataset was split into a daytime dataset (data points with a solar zenith angle $< 70°$) and a nighttime dataset (data points with a solar zenith angle $> 70°$) and were considered in separate models. Though two different models might lead to rough transitions between daytime and nighttime estimates, accurate estimates were in the foreground of this study, leading to the decision of separate models according to data availability. The response variables (rainfall yes/no and rainfall quantities) were taken from the rain gauge measurements.

The years 2010 to 2012 were used for model training. The year 2013 was used for validation. The retrieval process was two-step and consisted of (i) the identification of precipitating cloud areas and (ii) the assignment of rainfall quantities. All 2010 to 2012 data from the rain gauges that are masked as cloudy by the cloud mask products were used for training the rainfall area

model. All recorded rainfall events were used for training the rainfall quantities model. The resulting training dataset comprised 917774 (daytime) and 1409072 (nighttime) samples for the rainfall area training and 69703 (daytime) and 129325 (nighttime) samples for training of rainfall quantities from 26243 individual MSG scenes.

### 2.3.3 Tuning and model training

A single-hidden-layer feed-forward neural network was applied as machine learning algorithm. The spectral channels of MSG SEVIRI as well as the channel differences served as input nodes (predictor variables). The neural network was then applied to learn the relations between these spectral information and rainfall areas or rainfall quantities, respectively. In this context, a sophisticated pre-selection of input variables is not required, as the network is able to deal with correlated and even un- informative predictors unless their number is very high (Meyer et al., 2017), which was not the case in this study. For the
technical realisation, all steps of model training were performed using the R environment for statistical computing (R Core Team, 2016). The neural network implementation from the "nnet" package (Venables and Ripley, 2002; Ripley and Venables, 2016) in R was used in conjunction with the "caret" package (Kuhn, 2016) that provides enhanced functionalities for model training, estimation and validation.

Neural networks require two hyperparameters to be tuned to avoid under- or overfitting of the data: the number of neurons
in the hidden layer, as well as the weight decay. The neurons in the hidden layer represent nonlinear combinations of the input data and their number influences the performance of the model (Panchal et al., 2011). Weight decay penalizes large weights and controls the generalisation of the outcome (Krogh and Hertz, 1992). The number of neurons as well as weight decay were tuned using a stratified 10-fold cross-validation. Thus, the training samples were randomly partitioned into 10 equally sized folds with respect to the distribution of the response variable (i.e., raining cloud pixels, rainfall rate). Thus, every fold is a subset
(1/10) of the training samples and has the same distribution of the response variable as the total set of training samples. Models were then fitted by repeatedly leaving out one of the folds. The performance of a model was then determined by predicting on the held back fold. The performance metrics from the held back iterations were averaged to the overall model performance for the respective set of tuning values. For the rainfall areas classification models, the distance to a "perfect model", based on Receiver Operating Characteristics (ROC) analysis (see Meyer et al. (2016) for its application in rainfall retrievals) was used
as decisive performance metric. For the rainfall quantities regression models, the Root Mean Square Error (RMSE) was used. The number of hidden units were tuned for each value between two and the number of predictor variables. Weight decay was tuned between 0 and 0.1 with increments of 0.02 (Kuhn and Johnson, 2013). For training of rainfall areas, the threshold that separates rainy from non-rainy clouds according to the estimated probabilities was an additional tuning parameter. The optimal threshold was expected to be considerably smaller than 0.5 since the amount of non rainy samples was higher than the amount
of rainy samples. Therefore, the range of tested thresholds was 0 to 0.1 with increments of 0.01, and 0.4 to 1 with increments of 0.1. See Meyer et al. (2016) for further details of the threshold tuning methodology.

The optimal values for the hyperparameters that were revealed in the tuning study (Tab. 1) were adopted for the final model fitting. In this step, the model is fit to all training data using the optimal hyperparameters.

**Table 1.** Optimal hyperparameters for the individual models revealed during the tuning study and applied in the final model fitting.

|  | Number of neurons | Weight decay | Threshold |
|---|---|---|---|
| Rainfall areas at daytime | 5 | 0.05 | 0.07 |
| Rainfall areas at nighttime | 5 | 0.07 | 0.01 |
| Rainfall quantities at daytime | 5 | 0.05 | |
| Rainfall quantities at nighttime | 5 | 0.05 | |

### 2.3.4 Spatial estimations of rainfall

Final models were applied to all hourly MSG SEVIRI scenes from 2010-2014 for the Southern Africa extent to obtain spatio-temporal estimates of rainfall. Therefore, the clouded areas of a scene were first classified into rainy or not rainy using the respective model. The rainfall quantities were then estimated for the estimated rainfall areas. To ensure consistency within one
scene, the choice of the model being applied (either the daytime or nighttime model) was made according to the mean solar zenith angle of the respective scene. If the mean solar zenith angle was $< 70°$, rainfall for the entire scene was estimated using the daytime model. For scenes with a mean solar zenith angle $> 70°$, the nighttime model was applied.

### 2.4 Validation

Model estimates and weather station records from the entire year 2013 were used as independent data for model validation.
For the validation of estimated rainfall areas, all pixels at the location of the weather stations that were classified as cloudy by the cloud mask product were considered. Therefore the information from the weather stations about whether it was raining or not was compared to the model estimate for the respective MSG pixel. The validation data contained 403211 samples during daytime and 565415 samples during nighttime. Average hourly Probability Of Detection (POD), Probability Of False Detection (POFD), False Alarm Ratio (FAR) and Heidke Skill Score (HSS) were calculated as validation metrics. The POD gives the
percentage of rain pixels that the model correctly identified as rain (Tab. 2, 3). POFD gives the proportion of non-rain pixels that the model incorrectly classified as rain. The FAR gives the proportion of estimated rain where no rain is observed. The HSS also accounts for chance agreement and gives the proportion of correct classifications (both rain pixels and non-rain pixels) after eliminating expected chance agreement.

**Table 2.** Confusion matrix as baseline for the calculation of the verification scores used for the validation of the rainfall area estimates.

|  |  | Observation | |
|---|---|---|---|
|  |  | **Rainfall** | **No Rainfall** |
| **Estimation** | **Rainfall** | True positives (TP) | False positives (FP) |
|  | **No Rainfall** | False negatives (FN) | True negatives (TN) |

**Table 3.** Categorical metrics for validation of rainfall area estimates.

| Metric | Formula | Range | otimal value |
|---|---|---|---|
| Probability Of Detection | $POD = \frac{TP}{TP+FN}$ | 0 - 1 | 1 |
| Probability Of False Detection | $POFD = \frac{FP}{FP+TN}$ | 0 - 1 | 0 |
| False Alarm Ratio | $FAR = \frac{FP}{TP+FP}$ | 0 - 1 | 0 |
| Heidke Skill Score | $HSS = \frac{TP*TN-FP*FN}{[(TP+FN)*(FN+TN)+(TP+FP)*(FP+TN)]/2}$ | $-\infty$ - 1 | 1 |

To evaluate the ability of the model to estimate rainfall quantities, the correlation between the measured and the estimated hourly rainfall was calculated using Spearman's Product Moment Correlation (rho) to account for a non-normal distribution of the data. RMSE was also calculated. All cloudy data points (including non-rainy data points) were used for the validation of rainfall quantities. The rainfall quantities were further aggregated to daily, weekly and monthly rainfall sums to assess the performance of the model on different temporal scales.

## 2.5 Comparison to GPM

The results of the presented rainfall retrieval were compared to the rainfall estimates of the GPM mission. GPM, as a successor of the Tropical Rainfall Measuring Mission (TRMM), consists of an international network of satellites designed for worldwide high resolution precipitation estimates (Hou et al., 2014; Skofronick-Jackson et al., 2017). GPM provides data from March 2014 onwards. The GPM IMERG product estimates rainfall by combining all available passive-microwave estimates as well as microwave-calibrated infrared satellite estimates and data from rainfall gauges. GPM IMERG is available in 6h, 18h and 4 months latency.

In this study the 4 month latency (final product) with 30 minutes temporal and 0.1° spatial resolution (~10km x 10km) was used (Huffman et al., 2014). Due to different data availabilities of GPM IMERG, MSG as well as weather station data, the comparison was conducted for the overlapping time period late March 2014 to August 2014. GPM was aggregated from 30 minutes to 1h to match the temporal resolution of the MSG based estimates. Both products were validated using the weather station data as a reference. The performance metrics were compared between the MSG product and the GPM product on an hourly basis.

## 3 Results

### 3.1 Model performance

On average, 60 % of the rainfall observations were correctly identified as rainy by the model with a high number of scenes having much higher PODs (Fig. 3). The POFD was low (18 % in average) but the estimates featured a high FAR of 0.80. The average HSS per scene was 0.18. The POD was highest for high measured rainfall quantities and decreased for lower rainfall quantities (Fig. 4). FAR was highest for low predicted rainfall quantities and decreased for higher predicted quantities.

The average hourly RMSE was $0.72 \ mm \ h^{-1}$ (Fig. 5). Especially data points with low or medium measured rainfall could be estimated with low RMSE (Fig. 4). The RMSE was higher for high measured rainfall. Correlation indicated by Spearman's rho was 0.33 on hourly average. The performance of modeled rainfall quantities increased with the aggregation level (Fig. 6). The average correlation increased from rho = 0.33 (hourly) to 0.52 on a daily, 0.67 on a weekly and 0.71 on a monthly basis.

An overestimation of rainfall is observed especially when aggregated to monthly totals. An example of temporally aggregated rainfall estimates for 2013 are shown in Fig. 7.

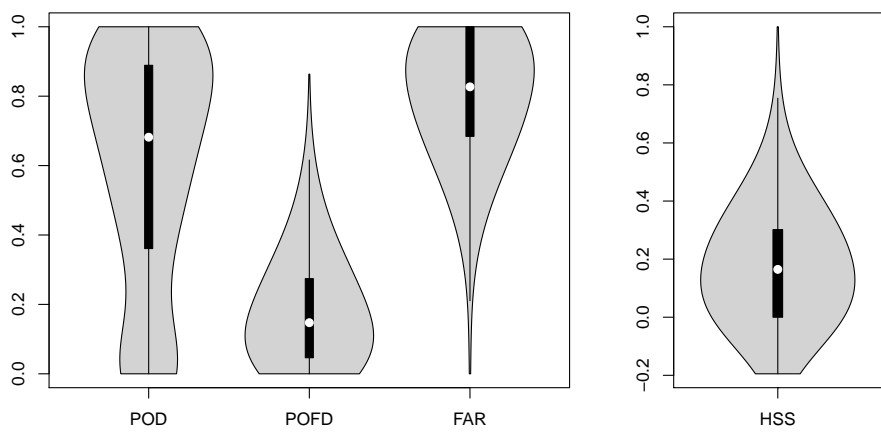

**Figure 3.** Validation of estimated rainfall areas for 2013 on an hourly basis. Each of the data points is the average performance of one hour. The data are visualized as "vioplot" where a boxplot is complemented by the kernel density of the data shown as grey areas at the sides of the boxplot.

## 3.2   Comparison to GPM

Compared to GPM IMERG, the MSG based rainfall retrieval for the period Mar-Aug 2014 showed a higher POD (0.57) than GPM IMERG (0.28) which considerably underestimated rainfall events (Fig. 8). In contrast, GPM IMERG had a lower FAR

(0.70) than the MSG based model (0.81). However, the FAR was high for both retrievals. The average HSS was the same for both retrievals (0.17), but the median HSS for GPM IMERG was 0 which was considerably lower than using the MSG based retrieval (0.10). Concerning the rainfall quantities, neither the correlation to measured rainfall nor the RMSE showed significant differences between both retrievals (Fig. 9). The average rho was 0.36 for the MSG based retrieval and 0.34 for GPM IMERG. The average RMSE was 0.88 for the MSG based retrieval and 0.85 for MSG IMERG.

Fig. 10 gives an example of the differences between the MSG based retrieval and GPM IMERG for 2014/04/24 12:00 UTC where severe floods occurred in the Eastern Cape province of South Africa. The colour composite of the corresponding MSG scene shows that clouds had a high optical depth in this area. The pattern is reflected in the estimates of the MSG based retrieval that estimated rainfall for the areas with high values of optical depth. This was partly confirmed by the weather station data.

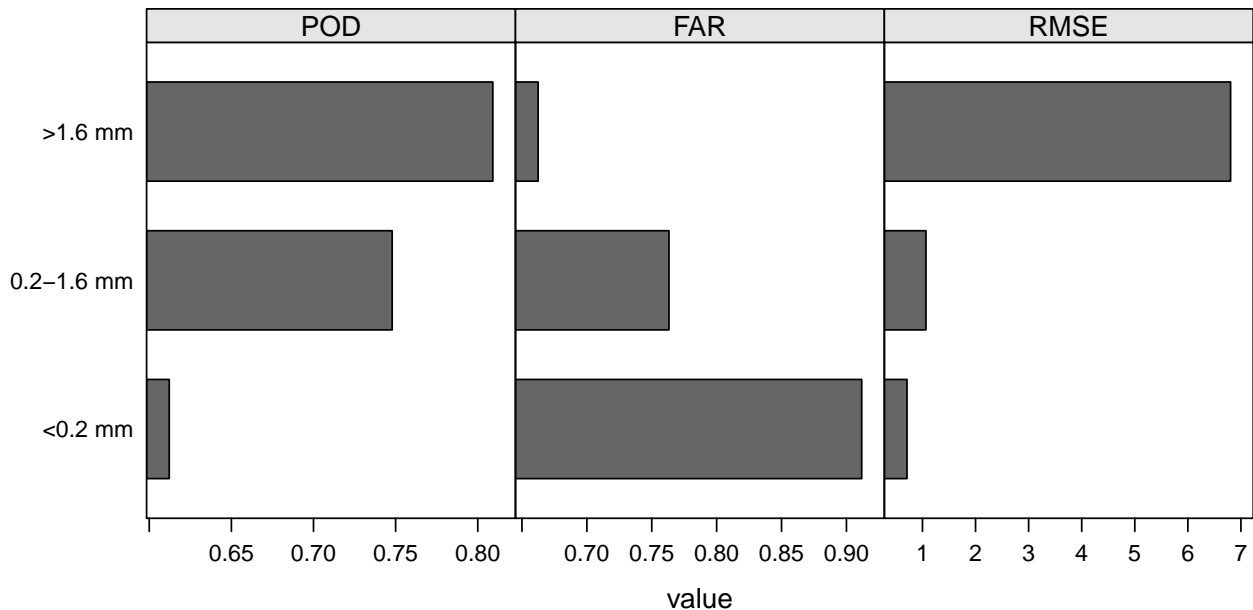

**Figure 4.** Comparison of POD for different hourly measured rainfall quantities as well as FAR for different predicted rainfall quantities. RMSE was compared for different measured rainfall quantities. All data points from 2013 were used for the calculation of the statistics. Thresholds for the three rainfall classes were set according to the first and third quartiles of the measured hourly rainfall quantities.

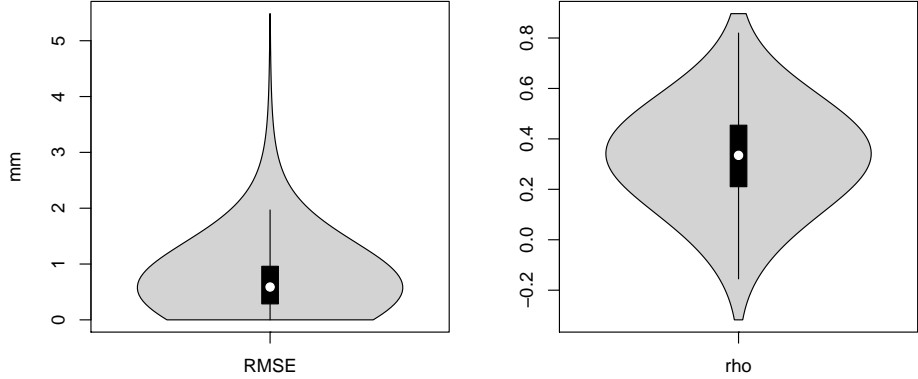

**Figure 5.** Validation of estimated rainfall quantities for 2013 on an hourly basis. Each of the data points is the average performance of one hour. See Fig. 3 for further information on the figure style.

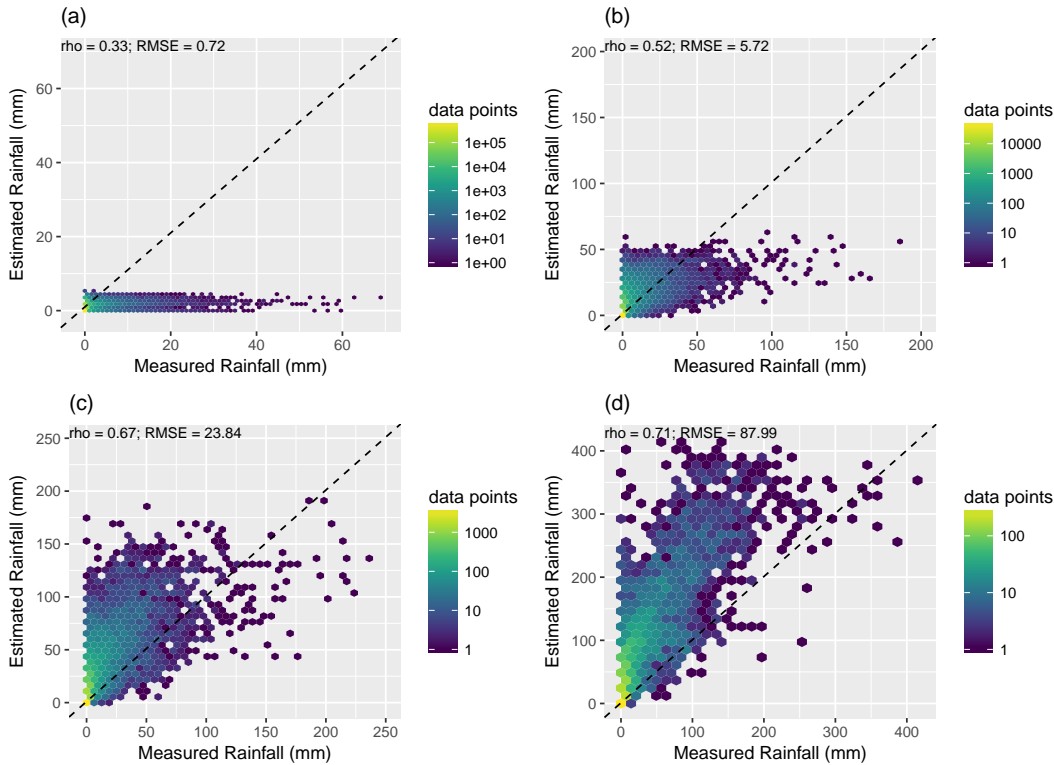

**Figure 6.** Validation of estimated rainfall quantities for 2013 at (a) hourly resolution on the different aggregation (b) daily, (c) weekly, (d) monthly. Each of the data points represents a station at the respective level of temporal aggregation. Rho represents the average correlation for each time step of the respective aggregation level. For an easy visual interpretation, the data are presented via hexagon binning where the number of data points falling in each hexagon are depicted by color.

However, rainfall was also estimated for areas where weather stations did not record any rainfall. In contrast, GPM IMERG showed an underestimation of rainfall areas, but still captured the high rainfall quantities that were recorded by the weather stations. The summary statistics for this hour are a POD of 0.75 for the MSG based retrieval and 0.19 for GPM IMERG. FAR was 0.65 and HSS 0.34 for the MSG based retrieval compared to a FAR of 0.89 and a HSS of 0.08 for GPM IMERG. The correlation between estimated and observed rainfall was 0.39 for the MSG based retrieval and -0.06 for GPM IMERG.

## 4    Discussion

The presented monthly maps reflect the general spatial and temporal rainfall patterns of Southern Africa as shown in Kruger (2007). They also reflect the annual characteristics of the year 2013. For example, the heavy rainfall events over southern Mozambique and the Limpopo River basin during mid January (Manhique et al., 2015).

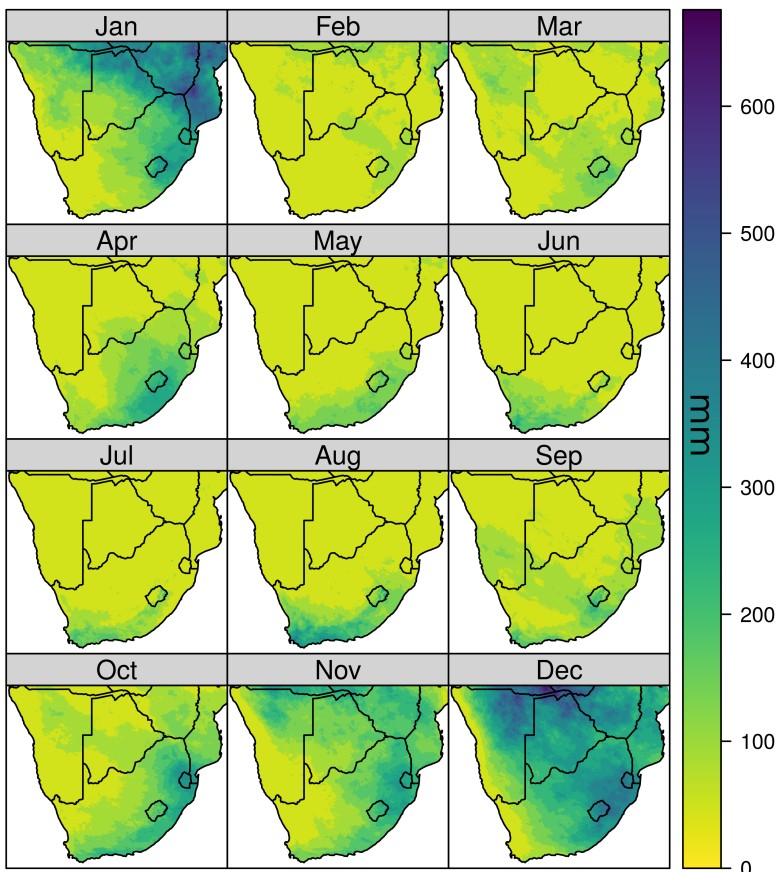

**Figure 7.** Monthly precipitation sums in mm of the year 2013 as estimated by this study.

The validation of the rainfall retrievals showed promising results but also highlights the difficulties of optical satellite-based rainfall estimates. The strength of the retrieval in terms of rainfall areas classification was a high POD for heavy rainfall events. The rainfall quantities for the heavy rainfall events were, however, underestimated in most cases. The major problem of the model was the overestimation of rainfall events leading to an overestimation of rainfall quantities. However, false alarms in the retrieval were generally predicted with low rainfall quantities. In this context, it is of note that in view to the scene-based validation strategy, FAR can easily increase in dry conditions when there are just a few false alarms in the estimates and no rainfall was observed by any station. However, the FAR was still high for hours with a considerable number of rainfall events. This might be partly explainable by spatial displacement due to parallax shifts. Though the shift is generally below 1 pixel in this region, even minor shifts can affect model training as well as the estimates. For future enhancement of the rainfall retrieval, a correction of the parallax shift (Vicente et al., 2002) would be appropriate. Differences in spatial and temporal scale are also an important issue especially since a majority of rainfall events in Southern Africa are of small spatial and temporal extent.

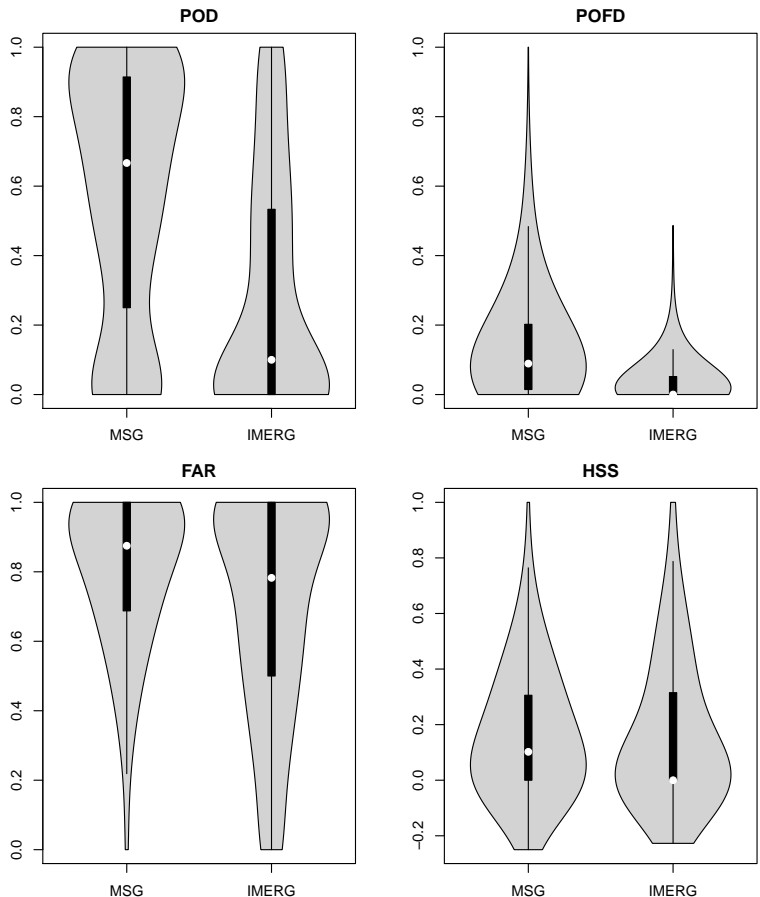

**Figure 8.** Comparison of the performance of the MSG based retrieval and GPM IMERG for rainfall area delineation between March and August 2014. Each of the data points is the average performance of one hour. See Fig. 3 for further information on the figure style.

The aggregation to an hour as well as the assumption that the weather station observation is representative for the entire pixel are also problematic, though essential. The issue of scale especially affects the broader resolution GPM IMERG data where a several km sized pixel is validated by a single point measurement. Beside of the issue of scale and spatial displacement, the retrieval technique depends on the quality of the rain gauge observations. Although the data was quality checked, common
5   problems associated with rain gauge measurements e.g. wind drift or evaporation leading to errors in the ground truth data and affect model training and validation remain (Kidd and Huffman, 2011). Also, due to different installation dates of the individual weather stations as well as the natural challenge of maintaining weather stations in remote areas, no gapless dataset could be compiled. Therefore, different sensor and data provider dependent calibration techniques, gaps in the time series of the data as well as the general problems associated with rain gauge measurements might lead to inconsistencies and uncertainties.
10   However, no reliable alternatives are available and rain gauge measurements are still considered as most reliable source of rainfall data.

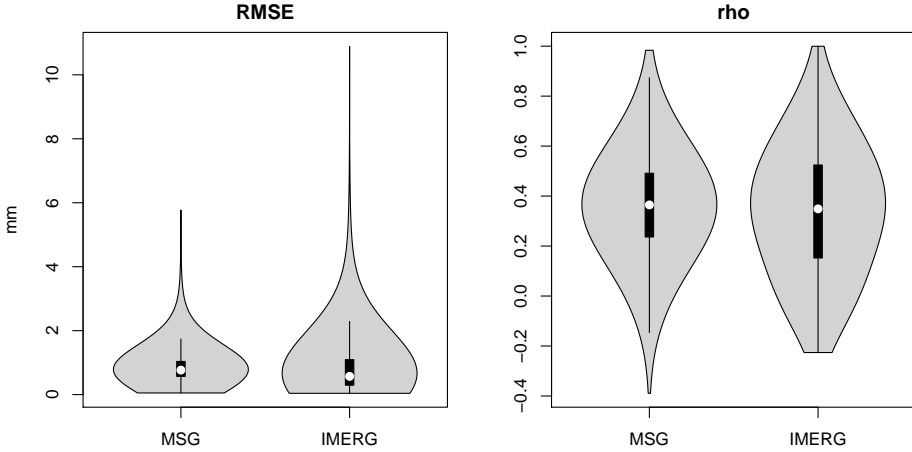

**Figure 9.** Comparison of the performance of the MSG based retrieval and GPM IMERG for hourly rainfall quantities between March and August 2014. Each of the data points is the average performance of one hour. See Fig. 3 for further information on the figure style.

The retrieval techniques relied on the cloud mask for an initial selection of relevant data points used for model training, validation and the final spatio-temporal estimates. Therefore, it can't be excluded that some data points were falsely excluded from the analysis as they were falsely masked as being not cloudy but rainfall was measured on the ground. However, we assume that rainy clouds are easy to capture by common cloud masking algorithms and that the resulting bias is therefore
comparably small.

Despite the errors and uncertainties associated with the presented rainfall retrieval, the combination of MSG data and neural networks are a promising approach. The model presented in this study outperformed the GPM IMERG product in terms of rainfall area detection where GPM IMERG considerably underestimated rainfall events. This behavior is partly explainable by scale because GPM IMERG has a coarser resolution of 0.1°. This makes local processes difficult to capture which is
an disadvantage considering that in Southern Africa especially small scale convective showers contribute to rainfall sums Kruger (2007). In terms of rainfall quantities, GPM IMERG and the presented retrieval did not show significant differences in correlation. The sample spatial comparison has shown that GPM IMERG has more differentiated rainfall estimates while the MSG based retrieval tends to estimate the mean distribution.

The presented MSG based retrieval is an easy to use method and allows for time series at a relatively high spatial resolution.
Aside of the promising results compared to GPM IMERG, the daily estimates of the MSG based retrieval are at least comparable to other products incorporated in the IPWG validation study IPWG (2016). A detailed comparison could currently not be given since validation data and strategy were not identical. Incorporation of the presented retrieval scheme to the IPWG validation study is intended by the authors for future assessment.

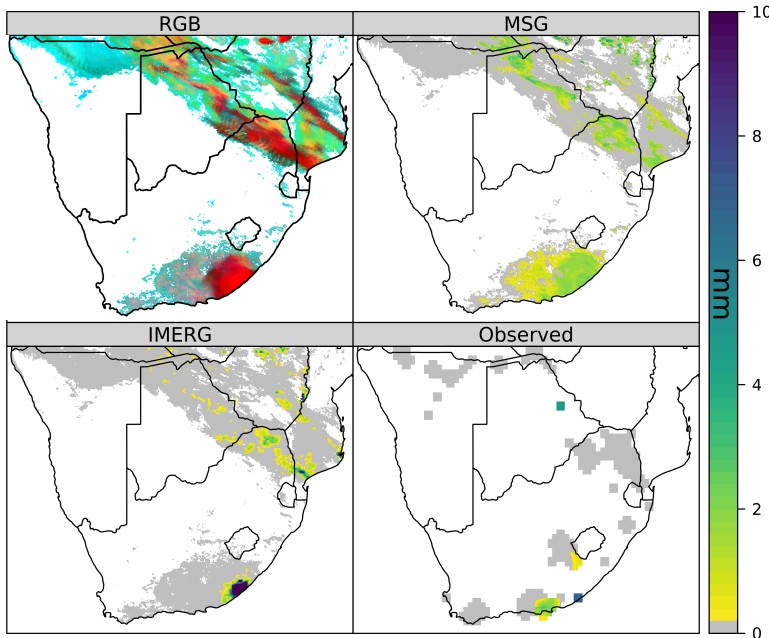

**Figure 10.** Sample satellite scene from 2014/04/24 10:00 UTC represented as a VIS0.8-IR3.9-IR10.8 false colour composite according to (Rosenfeld and Lensky, 1998) where cloud optical depth is indicated by red colouration, cloud particle sizes and phases in green and the brightness temperature modulates in blue. The rainfall estimates for this scene (estimated using the daytime model) are shown as well as the corresponding GPM IMERG product. Observed rainfall is depicted where weather station data were available. For visualization purposes, the spatial extent of the stations was increased. White background in the colour composite as well as in the MSG based retrieval and the GPM IMERG product represent no data due to missing clouds. In addition, white background in the representation of the observed rainfall is due to the absence of weather stations.

## 5 Conclusions

The rainfall retrieval technique developed in this study provides hourly rainfall estimates at high spatial resolution based on the spectral properties of MSG SEVIRI data and neural networks. The retrieval showed promising results in terms of rainfall area detection and estimation of rainfall quantities. However, the results also showed that the estimation of rainfall remains challenging. The main weakness of the presented retrieval was the overestimation of rainfall occurrence. However, the retrieval could compete with the GPM IMERG product in terms of rainfall quantity and was even better for rainfall area detection.

High resolution spatial datasets of rainfall is requested by a variety of research disciplines. The developed MSG based rainfall retrieval is able to deliver time series from the launch of MSG SEVIRI onward. An operationalization for near real-time rainfall estimates is intended. It can therefore serve as valuable dataset where high resolution rainfall for Southern Africa are needed. As an example it will serve as an important parameter within the "IDESSA" (An Integrative Decision Support System for Sustainable Rangeland Management in Southern African Savannas) project that aims to implement an integrative

monitoring and decision-support system for the sustainable management of different savanna types. The hourly and aggregated rainfall quantity estimations are available from the authors on request.

*Author contributions.* H. Meyer and T. Nauss designed the study. J. Drönner preprocessed the satellite data. H. Meyer developed the model code, performed the data analysis and prepared the manuscript with contributions from both co-authors.

5 *Competing interests.* The authors declare that they have no conflict of interest.

*Acknowledgements.* This work was financially supported by the Federal Ministry of Education and Research (BMBF) within the IDESSA project (grant no. 01LL1301) which is part of the SPACES-program (Science Partnership for the Assessment of Complex Earth System processes). We are grateful to the South African Weather Service for providing us with their rainfall data for South Africa and to SASSCAL WeatherNet for rainfall data from Namibia and Botswana. The cloud masking was done by using Level-2 data of the CLAAS-2 data record
10 provided by the EUMETSATs Satellite Application Facility on Climate Monitoring (CM SAF). The GPM IMERG V3 data were provided by the NASA/Goddard Space Flight Center's Mesoscale Atmospheric Processes Laboratory and Precipitation Processing System (PPS), which develop and compute the GPM IMERG V3 as a contribution to project GPM, and archived at the NASA GES DISC."

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
