# Peer review of "Satellite based high resolution mapping of rainfall over Southern Africa"

_Atmospheric Measurement Techniques, 2017_

## Referee Comment (RC1) · Anonymous Referee #1 · 21 Feb 2017

Review of Satellite based high resolution mapping of rainfall over Southern Africa by Hanna Meyer, Johannes Drönner, and Thomas Nauss.

General: The paper describes the formulation of a satellite-based precipitation estimation scheme based upon the MSG SEVIRI observations over southern Africa, and provides a comparison of this technique, together with that of the GPM IMERG product against gauge data. As such it is an interesting and useful paper since it covers a region that is often neglected.

My overall recommendation is that the paper is acceptable for publication following (minor/) major revision. The technical issues need to be addressed, in particular the ones relating to the masking of the data in the comparison (masking to just the MSG-

identified cloud regions could bias the statistics).

I would point the authors to the work of the International Precipitation Working Group team working on the South Africa data, also using gauge data to inter-compare daily precipitation products.

Key issues: i) Need to check the gauge data. First, ensure that the quality control is optimal, e.g. do some gauges never report rainfall? Do gauges distinguish between 'zero' and no-data? It is possible, once you have the satellite estimates, to check the performance of individual gauges – are there individual gauges that always are 'incorrect' compared to the satellite data? It would be unlikely that the satellite product would be consistently wrong over a particular gauge if it is correct over a neighbouring gauge.

ii) The use of the cloud mask in the statistical analysis (page xxx) removes regions where the gauge might report rainfall, but the satellite does not, thus, it biases the analysis.

iii) Although it is mentioned that the IMERG product is aggregated from the 30 minute product resolution to a 1-hour resolution, I could not find how the 15-minute MSG observations are aggregated into hourly estimates. Also, the authors should be careful with the time stamp of the products – do these relate to the start or end time (UTC) of the product? Also, is the gauge data in UTC or local time?

iv) (P6, first paragraph) Since there is a daytime and a nighttime 'algorithm', how do the two compare? In particular, since (presumably) the nighttime algorithm can be used both night and day, it could be used to assess the differences in performance. This is somewhat critical since a smooth transition in rainfall estimates between day and night is clearly desirable. Also, how do you define 'day' and 'night'?

General Technical issues: Check use of capitals for acronyms, e.g. P1, L3: 'Spinning Enhanced Visible and InfraRed Imager (SEVIRI)' Check the consistency of capitals,

e.g. P1, L6/7: '. . .(Probability of Detection, POD). However the False Alarm Ratio (FAR). . .'. Check use of acronyms: The general rule is, define all acronyms on first usage, after this only use the acronym (usually following on after the abstract). Only use an acronym if used more than once – and only if it is a commonly-used acronym (i.e. don't make up acronyms).

Specific Technical issues: P1, L1: consider 'necessary' instead of 'highly required' P1, L3 (and elsewhere): use of capitals for acronyms – 'Spinning Enhanced Visible and InfraRed Imager (SEVIRI)' P1, L4: remove 'for years' and replace 'truths' with 'truth' P1, L5: replace 'predicting' with 'the estimation of', and replace 'during' with 'over' P1, L6/7: '. . .(Probability of Detection, POD). However the False Alarm Ratio (FAR). . .'. P1, L10: Define 'IMERG' P1, L16: replace 'on a' with 'at' and replace 'resolution' with 'resolutions' P1, L20: replace 'An accurate' with just 'Accurate' P1, L21: replace 'in' with 'at' and 'resolution' with 'resolutions'

P2, L5: replace 'for entire' with 'covering the entire region of' P2, L11: replace 'resolution' with 'resolutions' and 'in' with 'at' P2, L12: replace 'can' with 'might'; insert 'would' after 'products'; insert 'degree of' before 'accuracy' and replace 'as' with 'since' P2, L16: replace ';' with 'and'; capitals for 'Meteosat Second Generation' and 'Spinning Enhanced Visible and InfraRed Imager' P2, L19: should 'South Africa' be 'southern Africa' (middle and end of line)? P2, L27: replace 'prediction' with 'estimation' P2, L30: replace 'yearly' with 'annual', remove 'sums' and replace 'follow' with 'follows'. P2, L32: replace 'rains' with 'rain'

P4, L1: replace 'sums' with 'totals' P4, L2: replace ';' with 'and'. P4, L5: remove 'the years' P4, L6: replace 'from' with 'at' P4, L7: remove 'the year'

P5, L4: The 3 x 3 km resolution is the IR resolution; i) the visible channels are about 1 x 1 km, but ii) the resolution over southern Africa for both the IR and visible channels is of course, poorer. P5, L9/10: the last sentence is gobbledygook: 'xx1 technology' if you google it, is to do with cycling, and the link to the web-page provided does not exist.

Reword/revise. P5, L11: remove 'the years' P5, L17: consider 'excluded' rather than 'masked' P5, L21: replace 'predict' with 'retrieve' P5, L25: replace 'many confusions' with 'much confusion' P5, L30: If all the channels are included in the NN, surely any channel differences should also considered within the NN without having to include them as separate entities?

P6, L7: replace 'two-folded' with 'two-step' (?) P6, last paragraph: see above regarding use of cloud mask, acronyms, use of capitals. P6, L34: the HSS can be bias-dependent since if all retrievals are zero and surface data non-zero, it will be dependent.

P7, L2: By 'Spearmans' I presume you mean the 'Spearman's Product Moment Correlation'; suugest rewording 'Spearmans rho' to 'Spearman's Product Moment Correlation (rho)' (or use the greek letter 'rho') P7, L2/3: replace 'Further the root mean square error (RMSE) was used' with 'The root mean square error (RMSE) was also calculated'. P7, L3: replace 'clouded' with 'cloudy' P7, L8: replace 'aiming at' with 'designed for' P7, L9: The reference to 'Smith et al., 2007' is somewhat antiquated: use 'Hou et al., 2014 and Skofronick-Jackson et al., 2017.' (Full references below) P7, L10: replace 'instruments' with 'estimates'

P8, L3: The initial sentence here is not evident from Figure 3. (see comments below about the box-plots). P8, L4: replace 'predictions' with 'estimates' P8, L5: presumably the '0.72 mm 4' should be '0.72 mmh-1' (use journal style for mm/hr) P8, L5: replace 'in' with 'on' P8, L6: reword 'rainfall quantities assignment' (I don't know what is meant by this). P8, L7/8: replace 'quantities could be' with 'is' P8, L8: replace 'rainfall sums' with 'totals' and 'predictions' with 'estimates' P8, L9: replace 'are show for the year 2013' with 'for 2013 are shown' P8, L30: replace 'Manhique et al. (2015).' with '(Manhique et al., 2015).'

P9, L1: replace 'retrieval' with 'retrievals' and replace 'highlights also' with 'also highlights' P9, L3: remove 'to elevated levels' P9, L5: parallax shifts would generally be < 1 pixel at this region.

P10, L3: move comma from after 'pixel' to after 'problematic' P10, L8: replace 'Kidd and Huffman (2001)' with '(Kidd and Huffman, 2011)' P10, L8/9: see comment above about checking gauge data. P10, L15: remove 'view to' P10, L16: replace 'GMP' with 'GPM'

P11, L2: Insert 'scheme' after 'retrieval' P11, L5: Insert 'technique' after 'retrieval' and replace 'in' with 'at'

P12, L1: 'overestimation of rainfall areas' – care is needed here – is there an over-estimation of 'rain area' or 'rain occurrence' (these are different, but linked). P12, L2: remove 'global'; remove 'assignment; replace 'even advantageous' with 'better' P12, L6: replace 'are' with'is'

References: Include data set references (most data sets now have doi's – and the GPM ones certainly do so).

Captions/Figures Figure 2: replace 'yearly' with 'annual'

Figure 3: replace 'predicted' with 'estimated'.

Figure 5: replace 'predicted' with 'estimated'; remove 'the year'; replace 'on' with 'at'; remove 'and on...levels'. Also, the colours seem to be smeared – particularly in (d) where each green point appears to be surrounded by a yellow 'ring'.

Figures 3,4,7 & 8: The box plots are not terribly good at conveying the necessary information. It would be much more valuable to display these a 'violin' plots (see Figure 5 of http://dx.doi.org/10.1175/JHM-D-16-0079.1)

Figure 9: would be good to include the gauge locations. Also, note that the MSG-estimate is a daytime retrieval scheme.

References: Hou, A. Y., and Coauthors, 2014: The Global Precipitation Measurements Mission. Bull. Amer. Meteor. Soc., 95, 701-722, doi:10.1175/BAMS-D-13-00164.1. Skofronick-Jackson, G., and Coauthors, 2017: The Global Precipitation Measurement

(GPM) Mission for Science and Society. Bull. Amer. Meteor. Soc., doi:10.1175/BAMS-D-15-00306.1, in press.

---

## Referee Comment (RC2) · Anonymous Referee #2 · 5 Mar 2017

Atmos. Meas. Tech. Discuss., doi:10.5194/amt-2017-33, 2017 Satellite based high resolution mapping of rainfall over Southern Africa By Hanna Meyer, Johannes Drönner, and Thomas Nauss

Anonymous Referee #2

The paper describes a method for estimating hourly rainfall over Southern Africa, based on a neural network approach, using MSG SEVIRI observations for the estimation and rain gauges data as ground truth. The results are compared to those obtained from IMERG of the Global Precipitation Measurement mission. The paper is interesting because it addresses an important and complex issue, as is the estimate of the surface precipitation in a region (the African continent) with sparse rain gauge and radar networks. I would like to recommend that this paper could be published after major/minor

revisions to address the following comments.

Major revisions: 1 – The description of some important aspects of the study is often done in a concise, not sufficiently complete and precise way to allow a direct and complete understanding. This fact is partly due to the use of some too general references (e.g. a conference (P2 L6 : IPWG, 2016) or books (P6, L13 : Venables and Ripley, 2002) or (P6, L17 : Kuhn and Johnson, 2013)), where more precise/accurate references (the paper in the conference or the section/pages in the books) would facilitate the understanding of the specific topics. In part it is due to the use of references that seem irrelevant/inconsistent with the text (P5, L8-9 : xxl technology . . .. OpenCL acceleration (see https://github.com/umr-dbs/xxl)). In part it is due to the use of specialized terms generally difficult to understand/interpret (P6, L15 : stratified 10-fold cross-validation). More attention to the aspects mentioned and a clearer description of the different topics would make it easier to read the text and would better highlight the most innovative aspects of the study.

2- Since the neural network is a key point in the study, more clarification on its design and its architecture would be appropriate. The references to texts (e.g. P6, L17 : Kuhn and Johnson, 2013) or packages (P6, L13 : "nnet" package (Venables and Ripley, 2002); P6, L14 : "caret" package Wing et al (2016)) do not lead to a direct understanding of the actual network used. The following points should be clarified: i) How the network input variables were selected (P5, L30 and P6, L1-2). The reference P6, L1 : Meyer et al. (submitted) is not available. ii) What is the network architecture (number of hidden levels and perceptrons) and how it has been designed. The text P6, l6-17 : The number of hidden units were tuned for each value . . .., is not clear in this regard. iii) What is the training procedure used in the study. Section 2.3.3 does not appear clear on this subject both for the language and the references provided (see point 1 above) and because the cited paper Meyer et al. 2016 does not provide more details about this procedure (apart from the threshold tuning methodology).

3 - The use of rain gauges as ground truth requires checks on the data quality. In the

paper some aspects of this issue should be developed, e.g check on no-data or no-rain, consistency between data from different networks. Is the retrieval quality depending on the rain gauges density?

4 – Figures 3 and 4 show the box plots concerning the POD, FAR, PDF, HSS, RMSE and rho evaluated considering the whole set of data; It would be more effective to evaluate these indexes considering different ranges of precipitation values (e.g. 0-25 mm, 25-50 mm etc).

Minor revisions:

1 – The section 2.2.2 should be modified by introducing a short description of the ability of the Seviri channels to provide information on the state of the atmosphere and the ground. This is important to clarify the choices that led to the selection of the neural network inputs.

2 – The performance of the retrieval technique (P8, L5-6) shown in fig. 5 (P11) could be presented in a more complete way by inserting in the four panels the corresponding RMSE and mean bias values. In the figure the colour bar (data point density) should be added.

3 – The reference to Smith et al. 2007 (P7, L9) can be updated with: Hou, A. Y., Kakar, R. K., Neeck, S., Azarbarzin, A. A., Kummerow, C. D., Kojima, M., Oki, R., Nakamura, K., and Iguchi, T.: The global precipitation measurement mission, B. Am. Meteorol. Soc., 95, 701-722, doi:10.1175/BAMS-D-13-00164.1, 2014.

4 – P6, L3 Please explain the criteria that has allowed to split the database into day and night.

5 – The paper contains a few typos that need to be corrected.

Please also note the supplement to this comment:
http://www.atmos-meas-tech-discuss.net/amt-2017-33/amt-2017-33-RC2-

supplement.pdf

---

## Author Comment (AC1) · 21 Apr 2017

Reviewer # 2
General: The paper describes the formulation of a satellite-based precipitation estimation scheme based upon the MSG SEVIRI observations over southern Africa, and provides a comparison of this technique, together with that of the GPM IMERG product against gauge data. As such it is an interesting and useful paper since it covers a region that is often neglected.
My overall recommendation is that the paper is acceptable for publication following (minor/) major revision. The technical issues need to be addressed, in particular the ones relating to the masking of the data in the comparison (masking to just the MSG-identified cloud regions could bias the statistics).

Response
Thank you very much for the comprehensive and detailed revision of our manuscript! In the following we would like to outline our response (green color) to your concerns (red color) as well as the subsequent changes that we made for the final version of the manuscript (green, italic).
* * *
I would point the authors to the work of the International Precipitation Working Group team working on the South Africa data, also using gauge data to inter-compare daily precipitation products.

Response
Yes, the IPWG does a lot of work on comparing different rainfall products for South Africa. We intend an incorporation of the presented retrieval to the IPWG validation study for future assessment and we are sure this would bring further insights into the strengths and weaknesses of the retrieval technique.
* * *
Key issues: i) Need to check the gauge data. First, ensure that the quality control is optimal, e.g. do some gauges never report rainfall? Do gauges distinguish between 'zero' and no-data? It is possible, once you have the satellite estimates, to check the performance of individual gauges – are there individual gauges that always are 'incorrect' compared to the satellite data? It would be unlikely that the satellite product would be consistently wrong over a particular gauge if it is correct over a neighbouring gauge.

Response
We totally agree that the quality of the ground truth data is an important issue! Yes, the data distinguish between zero and "no data" otherwise it would not be possible to train a model for rainy and non rainy clouds. All gauges that were used in this study provided rainfall for the training/testing period, however, with gaps. We added the following point to the discussion:

*"Also, due to different installation dates of the individual weather stations as well as the natural challenge of maintaining weather stations in remote areas, no gapless dataset could be compiled. Therefore, different sensor and data provider dependent calibration techniques, gaps in the time series of the data as well as the general problems associated with rain gauge measurements might lead to inconsistencies and uncertainties. However, no reliable alternatives are available and rain gauge measurements are still considered as most reliable source of rainfall data. "*

We further included information about the pre-processing of the data:

*"The data passed general provider-dependent quality checks before it was used in this study. This includes filtering of data beyond common data ranges, or situational checks for consistency with related parameters (e.g. air humidity) by SASSCAL. SAWS payed attention to rainfall values > 10 mm within 5 minutes and deleted those values if unreliable. Data from all providers was then included in an on-demand processing database system (Wöllauer et al., 2015) where it was automatically cross-checked for reliability by filtering values <0 and >500 mm rainfall per hour. All station data that provided sub-hourly information was aggregated to a temporal resolution of 1 hour within the database."*

We checked the results for station-dependent errors (see Figure below) and there were only few stations where the model constantly showed a low performance. We could identify 9 outliers that had a POD < 0.3 and a rho <0.05. The reason for this were large data gaps for the validation period so that the resulting statistics on a station basis are not meaningful for these stations. Only two stations showed abnormalities in the data that are probably associated with errors as they constantly featured very low rainfall values that were not at all in accordance with neighboring stations. These problems were not captured by the quality check methods. Therefore, it would be reasonable to remove such stations from the analysis. However, the problem is not critical for the results of this study. Since only very few stations (and very few data points in total) were affected, these data must be regarded as extreme outliers that have negligible effects.

[Figure]

ii) The use of the cloud mask in the statistical analysis (page xxx) removes regions where the gauge might report rainfall, but the satellite does not, thus, it biases the analysis.

Response
That is true! Our model relies on the cloud mask product that makes an initial selection of areas that come into question for rainfall. Honestly, we didn't think about this point as a potential source of error because we assume that rain clouds are easy to be captured as clouds by a cloud masking algorithm. To get an idea about the bias we visualized the fraction of data points for the year 2013 where this problem occurred: From 2108958 data points in total, roughly half of the data had no clouds and were not raining (1133885). 880071 data points were cloudy but it was not raining and 88555 data points were cloudy and it rained. It total, 6447 data points were not cloudy but it rained thus have to be regarded as problematic due to the cloud mask as initial selection (Figure A). This fraction is comparably small and if we compare the measured rainfall of those data points that were masked as cloudy with those that were not masked as cloudy (Figure B), the problematic data points had significantly lower measured rainfall, thus the problem luckily only slightly contributed to the rainfall totals.

Without going into detail with this analysis in the manuscript, we now accounted for this issue in the discussion section:
*"The retrieval techniques relied on the cloud mask for an initial selection of relevant data points used for model training, validation and the final spatio-temporal estimates. Therefore, it can't be excluded that some data points were falsely excluded from the analysis as they were falsely masked as being not cloudy but rainfall was measured on the ground. However, we assume that rainy clouds are easy to capture by common cloud masking algorithms and that the resulting bias is therefore comparably small."*

[Figure]

iii) Although it is mentioned that the IMERG product is aggregated from the 30 minute product resolution to a 1-hour resolution, I could not find how the 15-minute MSG observations are aggregated into hourly estimates. Also, the authors should be careful with the time stamp of the products – do these relate to the start or end time (UTC) of the product? Also, is the gauge data in UTC or local time?

Response

We agree that this information was missing in the manuscript. We now added the following information: "*To match the temporal resolution of all available rain gauge data, the extracted data were aggregated to hourly values. This was done by taking the median value of the four scenes available every hour. However, only if all four scenes were masked as cloudy, the corresponding hourly values for a respective station were used for further analysis. The extracted and aggregated MSG data were then matched with the corresponding rain gauge information under consideration of the time shift between MSG data (UTC) and rain gauge data (UTC + 2)*".

iv) (P6, first paragraph) Since there is a daytime and a nighttime 'algorithm', how do the two compare? In particular, since (presumably) the nighttime algorithm can be used both night and day, it could be used to assess the differences in performance. This is somewhat critical since a smooth transition in rainfall estimates between day and night is clearly desirable. Also, how do you define 'day' and 'night'?

Response

We now added the information about how the data were split into day and night:
"*Since the VIS and NIR channels of MSG are not available during the nighttime, the dataset was split into a daytime dataset (scenes with a solar zenith angle < 70°) and a nighttime dataset (scenes with a solar zenith angle > 70°)*"

We also added a section to describe how the spatial model estimates were created. Within one MSG SEVIRI scene, the model is used consistently as the mean solar zenith angle from the entire scene was used as

decisive angle:

*"Final models were applied to all hourly MSG SEVIRI scenes from 2010-2014 for the Southern Africa extent to obtain spatio-temporal estimates of rainfall. Therefore, the clouded areas of a scene were first classified into rainy or not rainy using the respective model. The rainfall quantities were then estimated for the estimated rainfall areas. To ensure consistency within one scene, the choice of the model being applied (either the daytime or nighttime model) was made according to the mean solar zenith angle of the respective scene. If the mean solar zenith angle was <70°, rainfall for the entire scene was estimated using the daytime model. For scenes with a mean solar zenith angle > 70°, the nighttime model was applied."*

We agree, the nighttime algorithm can be used for both, however, the VIS channels can increase the performance of the daytime models. In this context, we rated high performance for the daytime data being more important than smooth transitions. We added a short comment on that issue in the manuscript:

*"Though two different models might lead to rough transitions between daytime and nighttime estimates, accurate estimates were in the foreground of this study, leading to the decision of separate models according to data availability."*

The importance of the VIS channels is confirmed in by importance of the variables within the models. Though the importance of the variables within a neural network can only roughly be estimated, we certainly would yield a lower performance if the VIS information were not included.

[Figure]
* * *
General Technical issues: Check use of capitals for acronyms, e.g. P1, L3: 'Spinning Enhanced Visible and InfraRed Imager (SEVIRI)' Check the consistency of capitals, e.g. P1, L6/7: '. . .(Probability of Detection, POD). However the False Alarm Ratio (FAR). . .'.  Check use of acronyms: The general rule is, define all acronyms on first usage, after this only use the acronym (usually following on after the abstract). Only use an acronym if used more than once – and only if it is a commonly-used acronym (i.e. don't make up acronyms).

Response
We now defined all acronyms in the abstract (if they are used there) and then on the first appearance in the main text. We also made sure that the capitals are used consistently.
* * *
Specific Technical issues: P1, L1: consider 'necessary' instead of 'highly required' P1, L3 (and elsewhere): use of capitals for acronyms – 'Spinning Enhanced Visible and InfraRed Imager (SEVIRI)' P1, L4: remove 'for years' and replace 'truths' with 'truth' P1, L5: replace 'predicting' with 'the estimation of', and replace 'during' with 'over' P1, L6/7: '. . .(Probability of Detection, POD). However the False Alarm Ratio (FAR) . .'.P1, L10: Define 'IMERG' P1, L16: replace 'on a' with 'at' and replace 'resolution' with 'resolutions' P1, L20: replace 'An accurate' with just 'Accurate' P1, L21: replace 'in' with 'at' and 'resolution' with 'resolutions'

Response
We made all the suggested changes.
* * *
P2, L5: replace 'for entire' with 'covering the entire region of' P2, L11: replace 'resolution' with 'resolutions' and 'in' with 'at' P2, L12: replace 'can' with 'might'; insert 'would' after 'products'; insert 'degree of' before 'accuracy' and replace 'as' with 'since' P2, L16: replace ';' with 'and'; capitals for 'Meteosat Second Generation' and 'Spinning Enhanced Visible and InfraRed Imager' P2, L19: should 'South Africa' be 'southern Africa' (middle and end of line)? P2, L27: replace 'prediction' with 'estimation' P2, L30: replace 'yearly' with 'annual', remove 'sums' and replace 'follow' with 'follows'. P2, L32: replace 'rains' with 'rain' P4, L1: replace 'sums' with 'totals' P4, L2: replace ';' with 'and'. P4, L5: remove 'the years' P4, L6: replace 'from' with 'at' P4, L7: remove 'the year'

Response
Thank you, all changed.
* * *
P5, L4: The 3 x 3 km resolution is the IR resolution; i) the visible channels are about 1 x 1 km, but ii) the resolution over southern Africa for both the IR and visible channels is of course, poorer. P5, L9/10: the last sentence is gobbledygook: 'xx1 technology' if you google it, is to do with cycling, and the link to the web-page provided does not exist.

Response
Only the high resolution visible channel has a spatial resolution of 1 x 1 km. The "normal" visible channels still have a resolution of 3 x 3 km. We now accounted for the approx. resolution in southern Africa in our revised manuscript and made sure that we didn't use the high resolution channel:

*"MSG SEVIRI (Aminou et al. 1997) scans the full disk every 15 minutes with a spatial resolution of 3 x 3 km at sub-satellite point (~ 3.5 x 3.5 km in Southern Africa). Reflected and emitted radiances are measured by 12 channels, three channels at visible and very near infrared wavelengths (between 0.6 and 1.6 µm), eight channels ranging from near-infrared to thermal infrared wavelengths (between 3.9 and 14 µm) and one high-resolution visible channel with a spatial resolution of 1 x 1 km which was not considered in this study."*

We added more explanation about the processing scheme:
*"MSG SEVIRI Level 1.5 data (EUMETSAT 2010) were preprocessed to radiance values according to EUMETSAT (2012a) and brightness temperatures according to EUMETSAT (2012b) using a processing scheme based on a custom raster processing extension of the eXtensible and fleXible Java library (see https://github.com/umr-dbs/xxl) which enables parallel raster processing on CPUs and GPUs using OpenCL."*

However, we cannot understand your concern about the link to this web-page as we could reach this page via the link provided.
* * *
Reword/revise. P5, L11: remove 'the years' P5, L17: consider 'excluded' rather than 'masked' P5, L21:

replace 'predict' with 'retrieve' P5, L25: replace 'many confusions' with 'much confusion'

Response
changed
* * *
P5, L30: If all the channels are included in the NN, surely any channel differences should also considered within the NN without having to include them as separate entities?

Response
We agree since this is a point that we also discussed quite a bit. The predictor variables contains duplicated information in some way. However, we highly assume that these combinations are able to highlight patterns that are not obvious when only the individual channels are used. We found a significant increase of performance when the channel combinations were included which supports our assumption. Also, the neural networks are robust to duplicated information (at least considering comparably small numbers of predictor variables as we did in this study.), so that including this information is not of disadvantage for the model but allows taking advantage of highlighted patterns.

We now justified our decision in the manuscript:

*"Thus, the predictor variables contain the SEVIRI channels as well as channel combinations. Although this partially duplicates information, the channel combinations allow highlighting patterns that might not be apparent in the individual chan*nels."
* * *
P6, L7: replace 'two-folded' with 'two-step' (?) P6, last paragraph: see above regarding use of cloud mask, acronyms, use of capitals. P6, L34: the HSS can be bias-dependent since if all retrievals are zero and surface data non-zero, it will be dependent.

Response
We changed everything accordingly.
* * *
P7, L2: By 'Spearmans' I presume you mean the 'Spearman's Product Moment Correlation'; suugest rewording 'Spearmans rho' to 'Spearman's Product Moment Correlation (rho)' (or use the greek letter 'rho') P7, L2/3: replace 'Further the root mean square error (RMSE) was used' with 'The root mean square error (RMSE) was also calculated'. P7, L3: replace 'clouded' with 'cloudy' P7, L8: replace 'aiming at' with 'designed for' P7, L9: The reference to 'Smith et al., 2007' is somewhat antiquated: use 'Hou et al., 2014 and Skofronick-Jackson et al., 2017.' (Full references below) P7, L10: replace 'instruments' with 'estimates'

Response
We replaced the references and made all other changes as suggested
* * *
P8, L3: The initial sentence here is not evident from Figure 3. (see comments below about the box-plots). P8, L4: replace 'predictions' with 'estimates' P8, L5: presumably the '0.72 mm 4' should be '0.72 mmh-1' (use journal style for mm/hr) P8, L5: replace 'in' with 'on' P8, L6: reword 'rainfall quantities assignment' (I don't know what is meant by this). P8, L7/8: replace 'quantities could be' with 'is' P8, L8: replace 'rainfall sums' with 'totals' and 'predictions' with 'estimates' P8, L9: replace 'are show for the year 2013' with 'for 2013 are shown' P8, L30: replace 'Manhique et al. (2015).' with '(Manhique et al., 2015).'

Response

We made all required changes and the boxplots are changed to violinplots.

[Figure]

[Figure]
* * *
P9, L1: replace 'retrieval' with 'retrievals' and replace 'highlights also' with 'also highlights' P9, L3: remove 'to elevated levels' P9, L5: parallax shifts would generally be < 1 pixel at this region.

Response
Changed. We added the information that the parallax shift is rather small.
* * *
P10, L3: move comma from after 'pixel' to after 'problematic' P10, L8: replace 'Kidd and Huffman (2001)' with '(Kidd and Huffman, 2011)' P10, L8/9: see comment above about checking gauge data. P10, L15: remove 'view to' P10, L16: replace 'GMP' with 'GPM'
Response
Changed
* * *
P11, L2: Insert 'scheme' after 'retrieval' P11, L5: Insert 'technique' after 'retrieval' and replace 'in' with 'at'

Response
Changed
* * *
P12, L1: 'overestimation of rainfall areas' – care is needed here – is there an over-estimation of 'rain area' or 'rain occurrence' (these are different, but linked). P12, L2: remove 'global'; remove 'assignment; replace 'even advantageous' with 'better' P12, L6: replace 'are' with'is'

Response
We changed the wording to rain occurrence and made the other suggested changes
* * *
References: Include data set references (most data sets now have doi's – and the GPM ones certainly do so).

Response
SASSCAL and SAWS weather station data as well as the cloud mask products were all cited and

acknowledged in personal consultation with the providers and have no doi's. However, we now included more appropriate citation of GPM and MSG SEVIRI.

Huffman, G., Bolvin, D., Braithwaite, D., Hsu, K., Joyce, R., and Xie, P.: GPM L3 IMERG Late Half Hourly 0.1 degree x 0.1 degree Precipitation V03, Greenbelt, MD, Goddard Earth Sciences Data and Information Services Center (GES DISC). Accessed 15 June, 2015, doi:10.5067/GPM/IMERG/HH/3B, ftp://gpm1.gesdisc.eosdis.nasa.gov/data/s4pa/GPM_L3/GPM_3IMERGHH.03/, 2014.

EUMETSAT: High Rate SEVIRI Level 1.5 Image Data - MSG - 0 degree, http://navigator.eumetsat.int/discovery/Start/DirectSearch/DetailResult.do?f %28r0%29=EO:EUM:DAT:MSG:HRSEVIRI, 2010.
* * *
Captions/Figures Figure 2: replace 'yearly' with 'annual'; Figure 3: replace 'predicted' with 'estimated'.

Response
We changed both figure captions
* * *
Figure 5: replace 'predicted' with 'estimated'; remove 'the year'; replace 'on' with 'at'; remove 'and on. . .levels'. Also, the colours seem to be smeared – particularly in (d) where each green point appears to be surrounded by a yellow 'ring'.

Response
We improved the figure by providing a clearer binning of the values and a comprehensive color scheme with a legend showing the amount of data points.

[Figure]
* * *
Figures 3,4,7 & 8: The box plots are not terribly good at conveying the necessary information. It would be much more valuable to display these a 'violin' plots (see Figure 5 of *http://dx.doi.org/10.1175/JHM-D-16-0079.1*)

Response

Agreed. To take additional advantage of the data density, we changed the figure style to violin plots.
* * *
Figure 9: would be good to include the gauge locations. Also, note that the MSG-estimate is a daytime retrieval scheme.

Response
We included the note that the daytime scheme was used.
Including the location of the station turned out to be not really helpful: The relevant information is covered by the location of the stations, which is in our opinion, however, not very important since the location of all stations is already shown in Figure 1 and the location of the stations that recorded rainfall is shown in section d of this figure.
* * *
References: Hou, A. Y., and Coauthors, 2014: The Global Precipitation Measurements Mission. Bull. Amer. Meteor. Soc., 95, 701-722, doi:10.1175/BAMS-D-13-00164.1. Skofronick-Jackson, G., and Coauthors, 2017: The Global Precipitation Measurement (GPM) Mission for Science and Society. Bull. Amer. Meteor. Soc., doi:10.1175/BAMS-D-15-00306.1, in press.

Response
Thank you, both references are now included

---

## Author Comment (AC2) · 21 Apr 2017

Reviewer # 2

The paper describes a method for estimating hourly rainfall over Southern Africa, based on a neural network approach, using MSG SEVIRI observations for the estimation and rain gauges data as ground truth. The results are compared to those obtained from IMERG of the Global Precipitation Measurement mission. The paper is interesting because it addresses an important and complex issue, as is the estimate of the surface precipitation in a region (the African continent) with sparse rain gauge and radar networks. I would like to recommend that this paper could be published after major/minor revisions to address the following comments.

Response

Thank you very much for taking your time on our manuscript and your helpful comments! In the following we would like to outline our response (green color) to your concerns (red color) as well as the subsequent changes that we made for the final version of the manuscript (green, italic).
* * *
Major revisions:

1 – The description of some important aspects of the study is often done in a concise, not sufficiently complete and precise way to allow a direct and complete understanding. This fact is partly due to the use of some too general references (e.g. a conference (P2 L6 : IPWG, 2016) or books (P6, L13 : Venables and Ripley, 2002) or (P6, L17 : Kuhn and Johnson, 2013)), where more precise/accurate references (the paper in the conference or the section/pages in the books) would facilitate the understanding of the specific topics. In part it is due to the use of references that seem irrelevant/inconsistent with the text (P5, L8-9 : xxl technology .... OpenCL acceleration (see https://github.com/umr-dbs/xxl)). In part it is due to the use of specialized terms generally difficult to understand/interpret (P6, L15 : stratified 10-fold cross-validation). More attention to the aspects mentioned and a clearer description of the different topics would make it easier to read the text and would better highlight the most innovative aspects of the study.

Response

We changed/added the following information towards a comprehensive description of the applied methods:

1) References: IPWG refers to a website which is THE reference to compare different rainfall retrievals for South Africa (see also comment from Reviewer 1) and must be mentioned here. Venables and Ripley, 2002 refers to the software implementation being used. The nnet package is supposed to be cited in this way (see https://cran.r-project.org/web/packages/nnet/citation.html), though we agree it's a very general reference. We therefore added the direct reference to the software package:

*Ripley, B. and Venables, W.: nnet: Feed-Forward Neural Networks and Multinomial Log-Linear Models, http://CRAN.R-project.org/package=nnet, r package version 7.3-12, 2016.*

For Kuhn and Johnson we adapted the chapter and pages.

2) According to your suggestions we expanded the description of the input variables being used (see comment 2), the preprocessing of the satellite data, the architecture of the neural network (see comment 2ii) and the cross validation.

About the preprocessing of the data: *"MSG SEVIRI Level 1.5 data (EUMETSAT 2010) were preprocessed to radiance values according to EUMETSAT (2012a) and BBT values according to EUMETSAT (2012b) using a processing scheme based on a custom raster processing extension of the eXtensible and fleXible Java library (see https://github.com/umr-dbs/xxl) which enables parallel raster processing on CPUs and GPUs using OpenCL."*

About stratified 10-fold cross validation: *"Thus, the training samples were randomly partitioned into 10 equally sized folds with respect to the distribution of the response variable (i.e., raining cloud pixels, rainfall rate). Thus, every fold is a subset (1/10) of the training samples and has the same distribution of the response variable as the total set of training samples. Models were then fitted by repeatedly leaving out one of the folds. The performance of a model was then determined by predicting on the held-back fold. The performance metrics from the hold-out iterations were averaged to the overall model performance for the respective set of tuning values. For the rainfall areas classification models, the distance to a "perfect model", based on Receiver Operating Characteristics (ROC) analysis (see cite{Meyer2016} for its application in rainfall retrievals) was used as decisive performance metric. For the rainfall quantities regression models, the Root Mean Square Error (RMSE) was used."*
* * *
2- Since the neural network is a key point in the study, more clarification on its design and its architecture would be appropriate. The references to texts (e.g. P6, L17 : Kuhn and Johnson, 2013) or packages (P6, L13 : "nnet" package (Venables and Ripley, 2002); P6, L14 : "caret" package Wing et al (2016)) do not lead to a direct understanding of the actual network used. The following points should be clarified:

i) How the network input variables were selected (P5, L30 and P6, L1-2). The reference P6, L1 : Meyer et al. (submitted) is not available.

Response
Meyer et al. (submitted) was in review when this manuscript was submitted. It is now published in Remote Sensing Letters so we could include the correct reference:

Meyer, H.; Kühnlein, M.; Reudenbach, C. & Nauss, T.: Revealing the potential of spectral and textural predictor variables in a neural network-based rainfall retrieval technique. *Remote Sensing Letters,* **2017***, 8,* 647-656.

Concerning the choice of predictor variables: We agree that a  paragraph describing the general idea of the relation between MSG channels and cloud properties in section 2.2.2 is missing. We therefore added the following information:

*"The rainfall retrieval technique presented here works under the assumption that VIS, NIR and IR channels of MSG SEVIRI provide proxies for microphysical cloud properties, which are, in turn, related to rainfall. VIS and NIR channels have been shown to be related to cloud optical depth (Roebeling et al., 2006; Benas et al., 2017) and cloud water path (Kühnlein et al., 2014b) where the NIR channel is further related to cloud particle size (Roebeling et al., 2006). The IR channels have been shown to provide information about the cloud top temperature which was used as a proxy for cloud height (Hamann et al., 2014). The cloud droplet effective radius as well as liquid water path during night was approximated using IR channel differences (Merk et al., 2011; Kühnlein et al., 2014b)."*

From a technical perspective, it is no problem to insert all available information, even though individual channels might only have minor relations with rainfall. We added a note on that issue in the method section:

*"The function of the neural network is then to learn the relations between the spectral information and rainfall areas or rainfall quantities, respectively. In this context, a sophisticated pre-selection of input variables is not required, as the network is able to deal with correlated and even uninformative predictors unless their number is very high (Meyer et al., 2017), which was not the case in this study."*
* * *
ii) What is the network architecture (number of hidden levels and perceptrons) and how it has been designed. The text P6, l6-17 : The number of hidden units were tuned for each value ...., is not clear

in this regard.

Response
We made the architecture clear and improved the description of the hyperparameters that required tuning.

*"A single-hidden-layer feed-forward neural network was applied as machine learning algorithm. The spectral channels of MSG SEVIRI as well as the channel differences served as input nodes (predictor variables). The neural network was then applied to learn the relations between these spectral information and rainfall areas or rainfall quantities, respectively. In this context, a sophisticated pre-selection of input variables is not required, as the network is able to deal with correlated and even uninformative predictors unless their number is very high (Meyer et al., 2017), which was not the case in this study. For the technical realisation, all steps of model training were performed using the R environment for statistical computing (R Core Team, 2016). The neural network implementation from the "nnet" package (Venables and Ripley, 2002; Ripley and Venables, 2016) in R was used in conjunction with the "caret" package (Kuhn, 2016) that provides enhanced functionalities for model training, estimation and validation."*

*"Neural networks require two hyperparameters to be tuned to avoid under- or overfitting of the data: the number of neurons in the hidden layer, as well as the weight decay. The neurons in the hidden layer represent nonlinear combinations of the input data and their number influences the performance of the model (Panchal et al., 2011). Weight decay penalizes large weights and controls the generalisation of the outcome (Krogh and Hertz, 1992). "*

The actual number of neurons in the hidden layer results from the model tuning. We added a table showing the final model settings.

*"Optimal hyperparameters for the individual models revealed during the tuning study and applied in the final model fitting."*

|                                 | Number of neurons | Weight decay | Threshold |
|---------------------------------|-------------------|--------------|-----------|
| Rainfall areas at daytime       | 5                 | 0.05         | 0.07      |
| Rainfall areas at nighttime     | 5                 | 0.07         | 0.01      |
| Rainfall quantities at daytime  | 5                 | 0.05         |           |
| Rainfall quantities at nighttime| 5                 | 0.05         |           |
* * *
iii) What is the training procedure used in the study. Section 2.3.3 does not appear clear on this subject both for the language and the references provided (see point 1 above) and because the cited paper Meyer et al. 2016 does not provide more details about this procedure (apart from the threshold tuning methodology).

Response
Meyer et al. 2016 was included as a reference for the threshold tuning. We now made clear that the final step is fitting the model to all training data using the optimal set of hyperparameters:
*"The optimal values for the hyperparameters that were revealed in the tuning study (Tab. 1) were adopted for the final model fitting. In this step, the model is fit to all training data using the optimal hyperparameters."*

We also included a short paragraph on the spatial estimations (see also minor comment 4):

*"2.3.4 Spatial estimations of rainfall*
*Final models were applied to all hourly MSG SEVIRI scenes from 2010-2014 for the Southern Africa extent*

*to obtain spatio-temporal estimates of rainfall. Therefore, the clouded areas of a scene were first classified into rainy or not rainy using the respective model. The rainfall quantities were then estimated for the estimated rainfall areas. To ensure consistency within one scene, the choice of the model being applied (either the daytime or nighttime model) was made according to the mean solar zenith angle of the respective scene. If the mean solar zenith angle was <70°, rainfall for the entire scene was estimated using the daytime model. For scenes with a mean solar zenith angle > 70°, the nighttime model was applied."*

Thus, our description of the training procedure now contains the selection of predictor and response variables as well as their preprocessing, the network architecture, the model tuning and cross validation approach, the final fit of the models and the validation as well as spatial model estimations Please let us know if you still miss information.
* * *
3 - The use of rain gauges as ground truth requires checks on the data quality. In the paper some aspects of this issue should be developed, e.g check on no-data or no-rain, consistency between data from different networks. Is the retrieval quality depending on the rain gauges density?

Response
We totally agree that the quality of the ground truth data is an important issue! Yes, the data distinguish between zero and "no data" otherwise it would not be possible to train a model for rainy and non rainy clouds. We now included information about the pre-processing of the data:

*"The data passed general provider-dependent quality checks before it was used in this study. This includes for example filtering of data beyond common data ranges, or situational checks for consistency with related parameters (e.g. air humidity) by SASSCAL. The data was then included in an on-demand processing database system (Wöllauer et al. 2015) where it was automatically cross-checked for reliability by filtering values < 0 and > 500 mm of rainfall per hour. All station data that provided sub-hourly information was aggregated to a temporal resolution of 1 hour within the database."*

Unfortunately, we don't see a way to ensure a consistency between data from different networks: First of all, we can't analyze weather inconsistencies exist or not because this would require having stations from the different networks at exactly the same location which we don't have. Second, even if we had, how to correct for inconsistencies? There would probably be ways in areas where a high number of stations are available so that systematic inconsistencies in the data could be studied in a robust way. But this is not possible for our study. Our study relies on (comparably) sparse data from different sensors and provider-specific ways to process data. We don't see a way to check and/or correct for inconsistencies. However, we now accounted for this point in the discussion:

*"Therefore, different sensor and data provider dependent calibration techniques, gaps in the time series of the data as well as the general problems associated with rain gauge measurements might lead to inconsistencies and uncertainties. However, no reliable alternatives are available and rain gauge measurements are still considered as most reliable source of rainfall data. "*

It's further difficult to test if the retrieval quality depends on the gauge density. An obvious test would be to correlate the station density with the performances. However, that wouldn't lead to meaningful results since areas with many stations allow for a robust signal while areas with low density of stations don't because there are simply too few data.

However, we assume that different densities don't affect the model performance for the following reasons: We assume that the density would certainly have an effect if we trained on few scenes only, because then the current conditions of the areas with high station densities are highlighted. However since we trained on a long period, the model saw different conditions from dry to wet to learn from. Therefore it should be able to

learn the relations between cloud properties and rainfall without the risk of over-fitting towards areas with high station densities.
* * *
4 – Figures 3 and 4 show the box plots concerning the POD, FAR, PDF, HSS, RMSE and rho evaluated considering the whole set of data; It would be more effective to evaluate these indexes considering different ranges of precipitation values (e.g. 0-25 mm, 25-50 mm etc).

Response
Thank you for this comment! We now compared POD for different measured rainfall quantities and we compared FAR for different predicted rainfall quantities. The results give valuable information about the performance: high rainfall rates could be very well recognized as rainy clouds by the model. Though the model overestimated rainfall (high FAR), the predicted rainfall quantities for these false alarms were comparably low. We accounted for this in the results:

*"The POD was highest for high measured rainfall quantities and decreased for lower rainfall quantities (Fig. 4). FAR was highest for low predicted rainfall quantities and decreased for higher predicted quantities. [...] Especially data points with low or medium measured rainfall could be estimated with low RMSE (Fig. 4).*

And in the discussion:
*"The strength of the retrieval in terms of rainfall areas classification was a high POD for heavy rainfall events. The rainfall quantities for the heavy rainfall events were, however, underestimated in most cases. The major problem of the model was the overestimation of rainfall events leading to an overestimation of rainfall quantities. However, false alarms in the retrieval were generally predicted with low rainfall quantities."*

We presented the figure as barplot since a similar boxplot representation is not possible in this context: The boxplots base on the POD/FAR/etc on a scene basis, thus the data points of the boxplot could not be assigned to a unique rainfall quantity class.
POD can't be calculated when no rainfall is measured and FAR/POFD can't be calculated when rainfall is measured/no rainfall is predicted. Therefore, we can't make sense of a HSS for different rainfall classes since it bases on POD and FAR. For that reason we only compared POD and FAR. We didn't follow your suggestion of a class-based comparison for the correlation coefficient. For the correlation we want to know the model's ability to distinguish between low and high rainfall. When we only consider small parts of the gradient we would lose too much information.

[Figure]
* * *
Minor revisions:
1 – The section 2.2.2 should be modified by introducing a short description of the ability of the Seviri channels to provide information on the state of the atmosphere and the ground. This is important to clarify the choices that led to the selection of the neural network inputs.

Response
See major comment 2i
* * *
2 – The performance of the retrieval technique (P8, L5-6) shown in fig. 5 (P11) could be presented in a more complete way by inserting in the four panels the corresponding RMSE and mean bias values. In the figure the colour bar (data point density) should be added.

Response
We improved the figure by providing a clearer binning of the values and a comprehensive color scheme with a legend showing the amount of data points. We also added the RMSE in addition to rho.

[Figure]
* * *
3 – The reference to Smith et al. 2007 (P7, L9) can be updated with: Hou, A. Y., Kakar, R. K., Neeck, S., Azarbarzin, A. A., Kummerow, C. D., Kojima, M., Oki, R., Nakamura, K., and Iguchi, T.: The global precipitation measurement mission, B. Am. Meteorol. Soc., 95, 701-722, doi:10.1175/BAMS-D-13-00164.1, 2014.

Response
done
* * *
4 – P6, L3 Please explain the criteria that has allowed to split the database into day and night.

Response
We now added the information about how the data were split into day and night:

*"Since the VIS and NIR channels of MSG are not available during the nighttime, the dataset was split into a daytime dataset (data points with a solar zenith angle < 70°) and a nighttime dataset (data points with a solar zenith angle > 70°)"*

We also added a section of how the spatial estimation of rainfall were created because in this case, the mean solar zenith angle of the entire scene was decisive for the choice of the model:

*"Final models were applied to all hourly MSG SEVIRI scenes from 2010-2014 for the Southern Africa extent to obtain spatio-temporal estimates of rainfall. Therefore, the clouded areas of a scene were first classified into rainy or not rainy using the respective model. The rainfall quantities were then estimated for the estimated rainfall areas. To ensure consistency within one scene, the choice of the model being applied (either the daytime or nighttime model) was made according to the mean solar zenith angle of the respective scene. If the mean solar zenith angle was <70°, rainfall for the entire scene was estimated using the daytime model. For scenes with a mean solar zenith angle > 70°, the nighttime model was applied."*
* * *
5 – The paper contains a few typos that need to be corrected.

Response
We checked the manuscript again for typos.

---

## Referee Report (RR1)

RC#2 - Review of Satellite based high resolution mapping of rainfall over Southern Africa By Hanna Meyer, Johannes Drönner, and Thomas Nauss - Anonymous Referee #2, 09 May 2017

Atmos. Meas. Tech. Discuss., doi:10.5194/amt-2017-33, 2017 Satellite based high resolution mapping of rainfall over Southern Africa By Hanna Meyer, Johannes Drönner, and Thomas Nauss

Anonymous Referee #2

In the revised version of the manuscript " Satellite based high resolution mapping of rainfall over Southern Africa", all the recommendations have been addressed and the manuscript has been changed accordingly. The manuscript is now suitable for publication.